# A Framework to Guide the Implementation of Best Practice Clinical Learning Environments in Community General Practice: Australia

**DOI:** 10.3390/ijerph18041482

**Published:** 2021-02-04

**Authors:** Belinda O’Sullivan, Helen Hickson, Rebecca Kippen, Donna Cohen, Phil Cohen, Glen Wallace

**Affiliations:** 1Rural Clinical School, Faculty of Medicine, University of Queensland, Toowoomba 4350, Australia; 2General Practice Supervisors Australia, Bendigo 3550, Australia; research@gpsa.org.au (H.H.); glen.wallace@gpsa.org.au (G.W.); 3La Trobe Rural Health School, College of Science, Health and Engineering, La Trobe University, Bendigo 3550, Australia; 4School of Rural Health, Monash University, Bendigo 3550, Australia; rebecca.kippen@monash.edu; 5MEERQAT Pty Ltd., St Kilda, Melbourne 3182, Australia; donna@meerqat.com.au (D.C.); phil@meerqat.com.au (P.C.)

**Keywords:** doctors, general practitioners, physicians, health workers, workforce, learn, quality, framework, guide, environment, primary care, education

## Abstract

Clinical education/training is increasingly being expanded to community general practice settings (primary care clinics led by doctors). This plays an important role in developing a skilled “primary-care ready” workforce. However, there is limited information to guide the implementation of high-quality learning environments suitable for the range of general practices and clinical learners they oversee. We aimed to develop a consensus-based framework to address this. A co-design participatory action research method involved working with stakeholders to agree a project plan, collect and interpret data and endorse a final framework. As a starting point, an initial draft framework was adapted from an existing framework, the Best Practice Clinical Learning Environment (BPCLE) Framework. We gathered feedback about this from a national GP Supervisor Liaison Officer Network (SLON) (experienced GP clinical supervisors) during a 90-minute face-to-face focus group. They rated their agreement with the relevance of objectives and elements, advising on clear terminology and rationale for including/excluding various components. The resulting framework was refined and re-tested with the SLON and wider GP educational stakeholders until a final graphically designed version was endorsed. The resulting “GP Clinical Learning Environment” (GPCLE) Framework is applicable for planning and benchmarking best practice learning environments in general practice.

## 1. Introduction

There is an increasing body of research that indicates health workforce development should occur from the ground up, based on meeting the needs of the community [1]. With an ageing population in many countries, the World Health Organization (WHO) notes the demand for integrated management of chronic and complex care (including for conditions like diabetes, asthma, heart disease and depression) is increasing [2]. Preventing and managing such diseases will depend on the availability of regular primary healthcare workers practising at the appropriate scope [3]. Achieving this relies on training a skilled, high-quality primary care workforce that is capable of working at the level required, in community-based medically-led primary care clinics, which we call “general practices”. Developing such health workers, in turn, depends on high-quality learning in general practice settings, facilitated by a comprehensive approach to structuring the general practice learning environment. However, there is no evidence of published frameworks to guide this.

The number and diversity of general practices and learners they support requires that any frameworks be able to fit a wide context of implementation. Learners may include students of undergraduate medicine, nursing, allied health, pharmacy, dentistry and First Nations health, as well as junior doctors and vocational trainees learning to be general practitioners. While most undergraduate learners may spend only a few hours, days or weeks in general practice, junior doctors may spend several months at a time and general practice (GP) registrars (who may train for 3–4 years to become a qualified GP doctor) may be in one practice for several years to learn on the job [4]. The quality of the learning environment could have the most significant impact on this group. However, for all types of learners, a comprehensively structured learning environment is important to support seamless educational interactions, observations and mix of learning interventions [5].

Many general practices function as private businesses. In this situation, doctors—who often own the business—work closely with other staff such as practice managers, nurses and allied health personnel, concurrently managing learning and practice caseload and administration, in an integrated way. This gets busier as practices get smaller and community demand increases. In particular, in small rural areas, there are fewer doctors available to support distributed populations over large geographic catchments [6]. As such, resources for general practice learning need to be both relevant and efficient to use to avoid detracting from business and service demands.

Many countries are actively shifting clinical learning to community general practice settings [7]. In part, this is a response to the need to find enough clinical learning placements for the increasing numbers of health students. This is also done to promote better distribution of the workforce outside of metropolitan and hospital settings, to community and rural settings [8,9,10]. As a deliberate strategy, general practice placements may also support health worker learning about the continuity of patient care, developing patient relationships and forming connections to people and place [11,12]. This helps to develop health workers cognizant of the social and environmental influences on health.

There is some literature about the qualities of positive learning environments in general practice, but this is yet to be aggregated into a single framework. A structured learning environment with clear, learning outcomes is considered helpful [13]. Within the learning environment, communication and patient safety have been noted as important [14]. Motivated supervisors, able to utilise the whole context to promote learning, and who value learners and undertaking teaching and mentoring roles are also noted as relevant [15], along with learning infrastructure [16].

In some countries, certain professional colleges or universities may include competency frameworks and curriculum standards to guide learning in general practice [17,18]. However, these may not cover how to structure the learning environment. National clinical supervisor competencies also overlook this, except for addressing the development of a “safe environment” to minimise risk [19].

Another framework, the Best Practice Clinical Learning Environment (BPCLE) Framework [20], was developed for multiple health disciplines in 2008–2009, in the state of Victoria, Australia. It was based on a literature review and stakeholder consultation in the hospital sector [20]. This framework was subsequently validated in additional health disciplines and clinical training settings [20]. A statewide rollout of the framework to public health services occurred in 2013–2014, registered community health services (CHS) in 2015 [21] and a range of other health and human services in 2018–2019. It provides a useful foundation for developing a framework for general practice.

With this background in mind, this paper aims to describe the development of a co-designed framework to guide the implementation of best practice clinical learning environments in general practice.

## 2. Materials and Methods

### 2.1. Context of This Study

This work was led by a peak national organisation funded by the Australian Government, GP Supervisors Australia (GPSA), which advocates for and supports general practice supervision policy, resources and activity. GPSA was motivated by a desire to better support new supervisors and practices to plan and improve the quality of their learning environment, noting that there were no aggregate resources for this. Australia is a useful case study for this work as it is seeking to better distribute the medical workforce, including into community settings, under an emerging National Medical Workforce Strategy [22]. Australia has a national training program developing general practitioners by in-practice general practice placements over 3–4 years, led by two colleges and implemented by nine regional training organisations [4].

### 2.2. Conceptual Design

We set out to draw on the available evidence of theory and practice related to general practice learning and, in particular, gather input from a wide range of stakeholders with detailed knowledge of the dynamics of general practice learning. For this reason, we adopted a co-design participatory action research method [23]. Involving stakeholders at every stage of the process was considered an excellent way to corroborate and contrast findings and achieve a product able to cope with the heterogeneity of context and opinion and to achieve a uniformly agreed set of terminology. In turn, this method was considered relevant to promoting uptake and implementation of the framework, since engaging end-users in setting project goals, collecting data, interpreting findings and validating results is more likely to produce outputs that are grounded in achieving a practical objective and that are realistic to implement [24,25].

Our method involved four phases (Table 1) and had ethics approval (Monash no. 21388, ratified at The University of Queensland no. 2019002083/21388).

### 2.3. Phase 1: Agreeing a Plan

In terms of implementing the research, we had assembled a research team of mixed policy and research experience so as to assist combining robust research with contextual and stakeholder knowledge. The team included researchers with experience of developing and implementing the BPCLE Framework (D.C., P.C., G.W.) [21] as well as researchers with experience of studying GP supervision and primary care (R.K., H.H., B.O.), developing co-designed frameworks for primary care (B.O., R.K.) and with policy experience related to general practice (G.W., D.C., P.C., H.H. and B.O.).

Phase 1 involved developing a background paper based on a rapid review of the international literature, posing a justification and set of principles for the framework. These underpinned a project plan which was then discussed with the GPSA Board (consisting of up to ten general practitioners from different locations in Australia, with over 20 years’ experience of general practice teaching and learning). The background paper and plan was approved and applied to preparing a draft framework, as outlined below.

### 2.4. Phase 2: Drafting and Feedback on a Framework

Drafting the framework involved the research team applying the background paper to independently review the elements of the existing BPCLE Framework, considering its relevance to the range of general practice contexts. GPSA obtained a license from the state-based Victorian Department of Health and Human Services (DHHS) which owns the copyright to the BPCLE Framework, to use that framework and adapt it for the purpose of this project.

The research team considered how well each of the elements and objectives of the BPCLE Framework applied to all learning stages, health workers and general practice settings. This involved iterative input from all the research team, to revise the framework’s six elements, 28 objectives and their respective descriptions, until a final version was agreed for circulating for wider comment. The project background materials and draft framework was then provided to a national supervisor network (the Supervisor Liaison Officer Network or SLON) for discussion. The SLON is a group of approximately 25 experienced GP clinical supervisors involved in sharing information about general practice teaching and learning across Australia, including representatives from all states and territories. This group was invited to discuss the draft framework at a face-to-face meeting in September 2019. The 90-minute discussion was led by a trained facilitator (PC) and observed by other members of the research team (HH and GW). The SLON group was chosen as it included GPs who were supervisors of multiple health workers in general practice and who understood existing curriculum and expectations of teaching and learning within general practice. They also represented diverse gender, career experience, practice locations, states and territories, practice structure and functions.

Participants had the chance to review the document ahead of its discussion (the week prior) to support reflection and depth of response during the face-to-face session. All SLON members gave written informed consent to take part in the discussion. For the discussion, the facilitator asked participants to consider each element and objective statement, e.g., “Education is included in all aspects of planning”, and rate its relevance to general practice using an electronic sliding scale from 0 (not at all) to 100 (completely). Participants did this on their own electronic devices, using a common link.

Participants were also asked to provide feedback about the clarity and meaning of each element and objective, through open commentary facilitated during the meeting, with the goal of informing suitable wording. The ratings and the discussions of the entire session were recorded. Recordings were transcribed and the data and de-identified transcripts were provided to the wider research team for consideration. A broad content analysis was used to interpret and understand key themes based on iterative team discussions about the qualitative and quantitative data [26]. Through this process, subjective biases were managed and the breadth of interpretive analysis enabled more robust conclusions.

### 2.5. Phase 3: Refining and Re-Testing the Framework

Based on the data collected in Phase 2, the main findings were applied to refine the framework, including rearranging context and its wording where there was consensus from the research team. The revised version, which we called the General Practice Clinical Learning Environment (GPCLE) Framework, together with a summary of the outcomes of the consultation with the SLON, were tabled for discussion by the GPSA Board. This was done to ensure the project deliverables were fit to the original objectives and still addressing the Board’s goals.

A refined graphically designed version of the GPCLE Framework was subsequently circulated to a group of purposefully selected GP supervisors and general practice registrars (trainees learning on the job in general practice), practice managers, training organisations and bodies governing educational assessment, between December 2019 and March 2020. Participants received these documents and approximately 1–2 weeks later, participated in individual online meetings to provide more feedback. At these meetings, participants were asked some key questions, specifically:What are your general impressions of the GPCLE Framework?Is there anything missing from the framework?Is this applicable in your working/teaching context and/or how could this be facilitated?Is there any wording that could be clearer?Is there any other feedback you have?

The interviewer took notes during each meeting. From the notes, the main themes were summarised for discussion with the wider research team and used to agree how to further refine the framework. On the basis of this, the graphically designed framework was updated.

### 2.6. Phase 4: Refining and Checking for Validity

Following Phase 3, the refined framework was re-circulated to the members of the GPSA Board and the SLON in May 2020. They were asked whether the current version had captured their key issues and whether there was anything else they preferred to include or refine. On the basis of this, the research team undertook further iterative refinement, mainly involving slight modifications to wording, until consensus was achieved. A final version of the GPCLE Framework was produced, and this was agreed and endorsed by the SLON and the GPSA Board.

## 3. Results

The Phase 1 background paper identified a range of literature about the clinical learning environment covering multiple disciplines. A total of 62 articles were reviewed, including 38 from Australia and New Zealand, 2 from Canada, 1 from each of China, Denmark and Saudi Arabia, 5 from the Netherlands, 11 from the United Kingdom and 3 from the United States. A relationship was identified between aspects of the learning environment, such as supervisor skills, effective communication skills and learning resources, and learner outcomes. Whilst most of the literature aligned with existing best practice learning environment frameworks like the BPCLE Framework, there was no published material consolidating all aspects of high-quality learning environments in general practice, tailored to suit all health disciplines.

Based on the literature, we developed principles to guide the development of a GP-focused framework, namely that the framework should address all types of learners, in diverse general practice settings and comprehensively cover all the elements of a high-quality learning environment. Further, we intended for the framework to be efficient to use in a general practice context.

The Board approved the project plan, and thereafter, the draft GPCLE Framework was developed by amending the original wording of the BPCLE Framework to reflect the specific context of general practice. For example, terminology such as “the organisation” was changed to “the practice”. Wording was also adjusted to fit expectations of a general practice business model where supervision and learning needs to be achieved whilst also managing ongoing patient care, practice administration and economic productivity, often under a fee-for-service remuneration model. Otherwise, the six elements were left largely unchanged from the original BPCLE Framework. Overall, 25 individuals participated in Phase 2, of whom 21 completed the ratings (Table 2). Twelve were female and participants were from all states and territories and a range of practice sizes and locations.

Generally, participants seemed interested and engaged with the concept of the framework. The mean relevance rating was strong across six elements ranging between 0.74 and 0.88, while the mean relevance rating across 28 objectives varied more widely, between 0.47 and 0.87 (Table 2). There was some variability between average ratings for different elements/objectives per SLON member, ranging from 0.51 to 0.96. Four participants accounted for many of the lower ratings (average rating for these four participants: 0.51–0.63).

The participant discussion about these elements and objectives highlighted a range of contrasting themes, with some participants commenting on the current burden for general practices to absorb extensive policies, guidelines and reporting requirements whilst operating independent business models. They noted many stakeholders place expectations on their time (primary health networks, government, colleges, regional training organisations), indicating this framework needs to be short and resourceful for general practice, rather than complex and imposed through regulation. Given the number of stakeholders involved in general practice governance and learning arrangements, it was also thought necessary to clarify the expectations of practices to lead positive relationships with training partners. This included avoiding confusion when practices intersected with many other educational players who also share responsibility for communication and support.

There was also much discussion about the language of the framework and adapting terms to suit the diversity of general practice businesses and their context like location and caseload. In particular, participants wanted the framework to better reflect that in their small business setting, many practice staff—not just the clinicians—are involved in creating and maintaining a best practice learning environment. This was considered central to the feasibility of hosting learners. Participants also noted that learners (often coming to them from hospital settings) were not always well prepared and may require substantial orientation to the community primary care healthcare model and business expectations within general practice.

Aside from these high-level discussions, participants made comments relevant to specific objectives. For example, it was noted that education staff in general practice come with a diversity of skills and perspectives of the different aspects of real-world general practice (as both clinician and teacher). This means that general statements such as: “Clinical education staff are high-quality” would be difficult to interpret. The objective: “There are structured learning programs and assessment” was also identified as requiring amendment to align with the idea of learning in general practice happening in relation to the learner’s needs and the opportunities within the practice. This highlighted that the framework required flexibility to fit for issues such as the practice size and the supervisor’s special skills/interests, rather than being overly prescriptive. Similarly, the objectives “There are appropriate ratios of learners to patients” and “There are appropriate ratios of learners to educators” were also identified as irrelevant, given that practices with fewer GPs and fewer patients still considered themselves very able to offer rich learning opportunities. Atop of these findings, our research team recognised the need for more clarity around the communication expectations, including ensuring these reflect specific communication required in relation to achieving learning goals, rather than being expressed as general communication objectives.

In Phase 3, after the research team refined the framework based on the Phase 2 findings, the GPSA Board agreed that the graphically designed framework aligned with their expectations and endorsed further validation work by the research team. At this stage, feedback was gathered from 16 participants (8 female/8 male) via individual interviews, including 9 experienced GP supervisors, 4 GP registrars, a practice manager, a representative from an educational assessment body and a GP academic. By the conclusion of this process, saturation of feedback had been reached. Consistent themes were identified in relation to satisfaction with the end-product and areas for improvement. The main improvement that participants noted was to edit the document for brevity. A strong theme was appreciation of the graphically designed numbered elements depicted on a single page and the capacity to apply the framework within the general practice educational context:

“(It) connects really well with the guides that GPSA has and the RTOs—Regional Training Organisations; checklists and templates.”

Another noted:

“It fits in really well with our accreditation and we would like to use for accreditation quality improvement activity.”

Some participants noted that the framework could be used to build structured resources and voluntary quality improvement around the supervision effort in general practice. One mentioned it could be useful to: “develop some case studies that could be used to connect the objective with the resources”.

As a result of this feedback, the research team undertook further iterative editing.

In Phase 4, the GPSA Board and an additional 16 participants of the national SLON meeting endorsed the final framework. They noted that implementation may be enabled through connecting the framework with current practice accreditation processes, and also by encouraging practices to advocate the quality of their own learning environment by applying the framework for benchmarking and planning their own learning environment. At this stage, the terms “education” and “training” were noted to have different meanings, which were then checked for clarity by the research team.

Some disclaimers were also noted for the framework, including that it was for “all health clinical learners”, “it covered community-based (non-hospital) general practice learning, rather than general practice in the after-hours or hospital setting” and that it was a “guide only” to support these elements being considered “most of the time” within best practice general practice learning environments. Based on this, the research team worded an introduction to the framework. The snapshot version of the final GPCLE Framework is depicted in Figure 1 and has been published elsewhere in its expanded form [27].

A summary of the differences between the original BPCLE Framework and the GPCLE Framework is depicted in Table 3, showing the degree of tailoring to general practice and the reduced size and simplification of elements to support rapid uptake across a busy GP sector.

## 4. Discussion

Our research outlines a structured effort to develop a holistic framework to guide the development of best practice clinical learning environments in general practice. The resulting GPCLE Framework, comprising six elements and 19 objectives, is likely to have a place in supporting general practice, as it originated from a recognised need for more resources and it involved industry stakeholders at every step of the process. The framework is expected to be particularly applicable for new teaching practices, new supervisors and for benchmarking and planning quality improvement for existing practices. It is presented in simple, clear language, providing for snapshot rapid appraisal of the elements required for best practice learning in this setting. This was consistent with its intent and principles. This allows for widespread use, at a critical time when learning in primary care settings is expanding.

Our review of the literature did not identify similar published studies from other countries, although it is possible that other frameworks addressing high-quality learning environments in general practice are used by industry but have not been peer-reviewed. Other research promotes “systems thinking” in health services and development [28] which is aligned with the approach used in our project. Internationalising frameworks requires testing these in various countries and health systems, which will be an important next step for this project [29]. However, in the Australian context, the GPCLE Framework complements the BPCLE Framework to provide guidance on implementing high-quality learning environments that are tailored to the clinical setting (hospital or community/general practice).

Whilst tools and resources that enable general practice businesses to thrive are likely to be adopted, we also note there is a risk of selective uptake. This is because practices with poor-quality learning environments may be the least likely to self-identify as needing resources or guidance. Instead of singling out these practices, it would be beneficial to provide this resource as a guide for all practices, allowing them the chance to develop and advocate their own practice as a high-quality learning environment.

As such, we plan to promote the uptake and circulation of this framework, firstly at the practice level and secondly within general practice training providers and educators across diverse clinical disciplines. In line with the feedback we received, the framework has been designed to be as easy and appealing to use as possible. Aside from providing guidance to teaching practices, other potential applications include guiding organisations like GPSA and training organisations in the development of resources and exemplar stories to guide improvements under each framework element. The framework can also be linked with other resources related to practice accreditation as a supportive tool. This was noted as important by stakeholders involved in the co-design process.

A key learning from this project was that it was possible to use the BPCLE Framework as a useful basis for considering general practice needs. The main modifications that we found were necessary were adjusting the BPCLE terminology for the practice and reducing its complexity for pragmatic use within a business model.

It is important to note that our co-design process was mostly focused on stakeholders within the vocational medical training process, as the predominant learner group in general practice settings. We suggest that further research could be done to validate our framework for other disciplines and learner levels that make use of general practices for clinical education and training, including by seeking university and non-medical clinician input. Furthermore, the framework should be tested for applicability to other countries. Finally, while our consultations to inform the development of the GPCLE Framework included a range of input, we expect that wider implementation should be monitored to inform further refinement.

## 5. Conclusions

This paper describes a co-design action research project to develop, refine and validate a best practice clinical learning environment framework specifically tailored to the context of general practice, which we title the “GPCLE Framework”. The framework sets out six elements that define best practice in this context, namely: the practice values learning, best practice clinical care, a positive learning environment, an effective general practice–training provider relationship, effective communication processes and appropriate resources and facilities. Each element is further defined by clear objectives set out in brief format, providing the first conceptual framework to guide general practices to plan, establish and regularly appraise the quality of their learning environment for different types of learners, regardless of their context.

## Figures and Tables

**Figure 1 ijerph-18-01482-f001:**
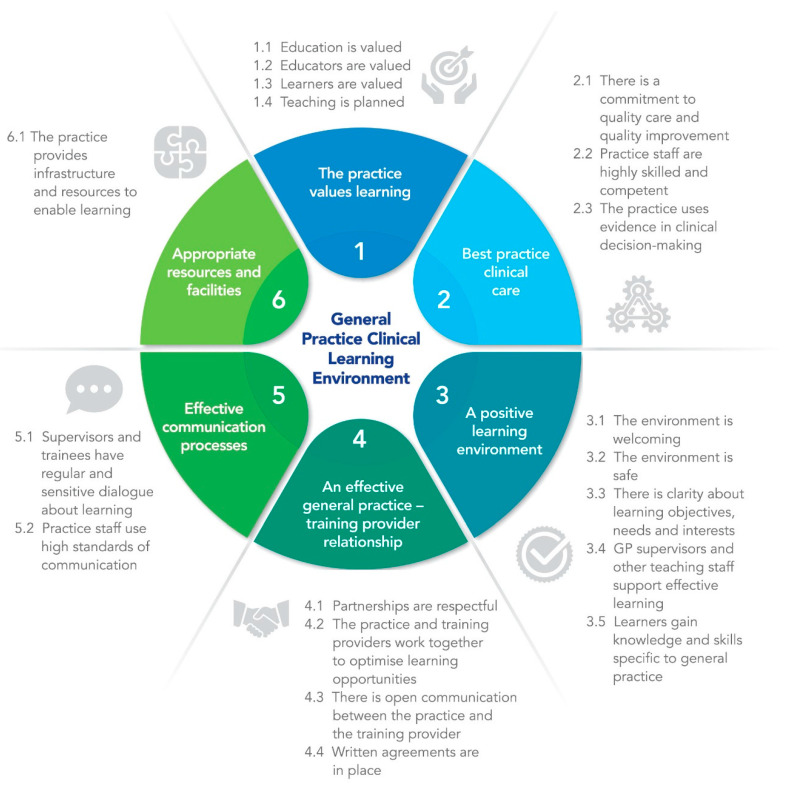
Final elements and objectives of the General Practice Clinical Learning Environment (GPCLE) Framework.

**Table 1 ijerph-18-01482-t001:** The four phases of the project.

Phases	Key Tasks
1: Agreeing a plan	Background literature appraisal Develop plan Agree principles for framework
2: Drafting and feedback on a framework	Draft GPCLE Framework ^a^ Circulate to a national Supervisor Liaison Officer Network Discussion face-to-face for 90 min, participants rating relevance of the material to General Practice
3: Refining and re-testing the framework	Refine framework and seek feedback from 16 participants via individual interviews.
4: Refining and checking for validity	Final version verified by GPSA Board and an additional 16 participants of the national Supervisor Liaison Officer Network. ^a^

^a^ GPCLE—General Practice Clinical Learning Environment; GPSA—General Practice Supervisors Australia.

**Table 2 ijerph-18-01482-t002:** Mean relevance rating scores for the proposed elements and objectives, as nominated by Supervisor Liaison Officer Network (SLON) focus group participants (Phase 2) ^a^

Element	Mean Rating	Objective	Mean Rating
An organisational culture that values learning	0.85	Education is valued	0.87
		Educators are valued	0.83
		Students/Learners are valued	0.83
		There is a career structure for educators	0.47
		Education is included in all aspects of planning	0.60
		Use of facilities and resources are optimised for all educational purposes	0.55
Best practice clinical practice	0.88	There is an organisational commitment to quality of care and continuous quality improvement	0.81
		Clinical staff are highly skilled, knowledgeable and competent	0.79
		The organisation adopts best evidence into practice	0.79
A positive learning environment	0.83	The environment is welcoming	0.86
		The environment is safe	0.87
		Appropriate learning opportunities take place	0.86
		There is clarity about educational objectives	0.77
		Clinical education staff are high-quality	0.81
		Learners are well prepared	0.63
		There are appropriate ratios of learners to educators	0.75
		There are appropriate ratios of learners to patients/clients	0.82
		There is continuity of learning experience	0.78
		There are structured learning programs and assessment	0.65
An effective GP practice–education provider relationship	0.80	Open communication occurs at all levels of the partner organisations	0.81
		Mutual respect and understanding exists between the health service and its training provider partner	0.80
		The partners assist each other to optimise their contribution to the training of health professionals	0.76
		Relationship agreements codify expectations and responsibilities of the partners in the delivery of clinical education	0.77
Effective communication processes	0.85	Communication is not taken for granted by the organisation	0.84
		Communication informs actions, behaviours and decision-making	0.85
		Communication facilitates feedback	0.84
		Communication facilitates improved teaching and learning	0.86
Appropriate resources and facilities	0.74	Learners and staff have access to the facilities and materials needed to optimise the clinical learning experience	0.82

^a^ Elements and objectives are presented as they were provided for discussion. Mean ratings for elements and objectives were considered within one pool (elements and objectives not separated). GP—general practice.

**Table 3 ijerph-18-01482-t003:** Alignment of the original Best Practice Clinical Learning Environment (BPCLE) Framework and the final version of the GPCLE Framework developed during the project. Objectives under each element follow the order used in the GPCLE version ^a^.

BPCLE Framework	GPCLE Framework
**1. An organisational culture that values learning**	**1. The practice values learning**
Education is valued	Education is valued
Educators are valued	Educators are valued
Learners are valued	Learners are valued
Education is included in all aspects of planning	Teaching is planned
There is a career structure for educators	
Use of facilities and resources are optimised for all educational purposes	
**2. Best practice clinical practice**	**2. Best practice clinical care**
There is an organisational commitment to quality of care and continuous quality improvement	There is a commitment to quality care and quality improvement
Clinical staff are highly skilled, knowledgeable and competent	Practice staff are highly skilled and competent
The organisation adopts best evidence into practice	The practice uses evidence in clinical decision-making
**3. A positive learning environment**	**3. A positive learning environment**
The environment is welcoming	The environment is welcoming
The environment is safe	The environment is safe
There is clarity about educational objectives	There is clarity about learning objectives, needs and interests
Clinical education staff are high-quality	GP supervisors and other teaching staff support effective learning
Appropriate learning opportunities take place	Learners gain knowledge and skills specific to general practice
Learners are well prepared	
There are appropriate ratios of learners to educators	
There are appropriate ratios of learners to patients	
There is continuity of learning experiences	
There are structured learning programmes and assessment	
**4. An effective health service-training provider relationship**	**4. An effective general practice–training provider relationship**
Mutual respect and understanding exists between the health service and its training provider partner	Partnerships are respectful
The partners assist each other to optimise their contribution to the training of health professionals	The practice and training providers work together to optimise learning opportunities
Open communication occurs at all levels of the partner organisations	There is open communication between the practice and the training provider
Relationship agreements codify expectations and responsibilities of the partners in the delivery of clinical education.	Written agreements are in place
**5. Effective communication processes**	**5. Effective communication processes**
Communication facilitates improved teaching and learning	Supervisors and trainees have regular and sensitive dialogue about learning
Communication is not taken for granted by the organisation	Practice staff use high standards of communication
Communication informs actions, behaviours and decision-making	
Communication facilitates feedback	
**6. Appropriate resources and facilities**	**6. Appropriate resources and facilities**
Learners and staff have access to the facilities and materials needed to optimise the clinical learning experience	The practice provides infrastructure and resource to enable learning

^a^ BPCLE— Best Practice Clinical Learning Environment. GPCLE—General Practice Clinical Learning Environment. The colours reflect the same colours for the elements used in Figure 1.

## Data Availability

The data are available upon request to the authors, subject to ethical approval.

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
