# Peer review of "A Framework to Guide the Implementation of Best Practice Clinical Learning Environments in Community General Practice: Australia"

_ijerph, 2021, doi:10.3390/ijerph18041482_

Round 1

Reviewer 1 Report

The aricle presents a complex process adapting with minor modifications the "Best Practice Clinical Learning Environment" (BPCLE) Framework to the context of General Practice (GP).

The resultant "GP Clinical Learning Environment" (GPCLE) Framework is a significant contribution to GP training, simplifying and harmonizing BCPLE to fullfil the GP needs.

Therefore the article could be published provided the authors will improve the description of the four phases of the project and include flowcharts , as their description of this process is complicated and difficult to follow. 

Author Response

The article presents a complex process adapting with minor modifications the "Best Practice Clinical Learning Environment" (BPCLE) Framework to the context of General Practice (GP).

Thank you for these positive comments

The resultant "GP Clinical Learning Environment" (GPCLE) Framework is a significant contribution to GP training, simplifying and harmonizing BCPLE to fulfil the GP needs.

Thank you for these positive comments

Therefore the article could be published provided the authors will improve the description of the four phases of the project and include flowcharts, as their description of this process is complicated and difficult to follow. 

To respond to this we added a new Table 1 at p.4 line 170-174 to describe the 4 phases. Then renumbered remaining tables.

Reviewer 2 Report

Introduction

In my opinion the introduction is too long and some paragraphs could be separated. The introduction section requires a better organization to keep a sequence in the explanation. A reorganization of certain paragraphs is necessary. For example, the second paragraph beginning with the sentence (line 78) "General practices of various sizes ..." should be placed before. 

Some of the comments included in the introduction section can go to the Discussion section.

The aims of the paper “This paper aims to describe… (line 55) should go to the end of the introduction, after commenting on the background of the topic and the justification for the study.

Materials and methods

The Materials and Methods section is detailed and rigorous. However, the number and characteristics of the participants should be detailed.

Results

In this section, there are many comments that, in my opinion, should be included in the discussion.

In the second paragraph (line 273, the results included must refer to Table 1 and must be included in the text at the end of the paragraph.

Discussion

In this section, it is necessary to discuss the paper with other studies. No references are included.

Author Response

 In my opinion the introduction is too long and some paragraphs could be separated. The introduction section requires a better organization to keep a sequence in the explanation. A reorganization of certain paragraphs is necessary. For example, the second paragraph beginning with the sentence (line 78) "General practices of various sizes ..." should be placed before. 

In response, we edited the introduction for brevity and clarity. See tools track changes throughout introduction, particularly at p.2, line 74 and 80.

We also added three paragraph breaks which must have jumped out when the paper was formatted:

p.2 line 62

p.3 line 110

p.3 line 120

Some of the comments included in the introduction section can go to the Discussion section.

We reviewed the introduction and decided not to move material to the Discussion. This was because after editing the introduction for brevity, we deemed it included a relevant range of background material to provide the reader with an understanding of the problem, what is known about it, why this is an important topic and what solving this might entail.

The aims of the paper “This paper aims to describe… (line 55) should go to the end of the introduction, after commenting on the background of the topic and the justification for the study.

In response we moved the Aim to the end of the introduction at p.3 line 138, and checked Introduction to ensure that this flows well.

Materials and methods

The Materials and Methods section is detailed and rigorous. However, the number and characteristics of the participants should be detailed.

Thank you for noting this. In response, we checked that we had included any numbers and gender mix/ distribution of the various participants in each of the Phases in the main text of the results. It is our view that the respondents should be reported in the Results section as they are an outcome of our recruitment strategy for doing the ratings (in Phase 2) and refining the framework (in Phase 3) and the validation work (in Phase 4). Because there were multiple stages and methods of data collection, the characteristics of participants were not reported in a single table.

Results

In this section, there are many comments that, in my opinion, should be included in the discussion.

Thank you for this suggestion. We reviewed the Results to check for this and we found that the results are definitely all results of the participatory action research that we did. This includes a mix of quantitative and qualitative findings from the four Phases of developing the GPCLE Framework. We then reviewed the Discussion to check that we adequately reflected on the findings and the implications only, and not the results. Minor edits were made as shown in tools track changes.

In the second paragraph (line 273, the results included must refer to Table 1 and must be included in the text at the end of the paragraph.

Thank you for noting this. Our tables were renumbered as we added a new Table 1 to highlight the four Phases of the project, (in response to another reviewer). Then we ensured that all tables were noted to in the text where the information about them was first mentioned and correctly numbered.

Discussion

In this section, it is necessary to discuss the paper with other studies. No references are included.

Thank you for suggesting this. In response, we added to the Discussion other references to show that we are aware of frameworks for global health systems development, and more clearly noted that there are no other existing frameworks in this field at p.13 line 463.

Reviewer 3 Report

This manuscript describes the development of the GPCLE-framework, based on an earlier version (the BPCLE-framework). The framework describes desirable elements and objectives for effective workplace learning environment. As such it can serve as a guideline for analysing and optimizing such learning environments.

The manuscript is well-written and describes the steps in developing the framework in a clear way. My main suggestion for improving the manuscript is that it should be made clear that the framework, and the way it was established, is relevant for the Australian context only. Aspects of the framework may be generic and/or applicable to other parts of the world, but in my opinion the framework as such cannot be applied elsewhere in unmodified form due to local characteristics of community practices and roles of health care professionals. I therefore suggest that the Title is completed by adding “… in Australia” (at the end of the Title).

Apparently, a literature-based background paper was developed in phase 1 (lines 169-170), but the content of this background paper is not shared or published independently. It would have been interesting to summarize at least the main findings of this study in order to see in which elements the GPCLE-framework can be considered generic (wider application) and in which elements it is specific for the Australian situation. Would it be possible to include the main findings of the background paper in this manuscript, in summarized form or table-format? Otherwise, at least a short referral to non-Australian literature would be appreciated. Also, in the Discussion section a short comparison to literature from other contexts is welcomed (lines 428-438) to make the manuscript more interesting to a non-Australian audience.

Other, minor remarks:

  1. The range of learners in general practice settings (lines 60-62) seems not to include pharmacists (unless the fall in the category allied health). Are pharmacists excluded deliberately or as a consequence of local situations? This seems strange as a tendency to increasing inter-professional practice and education is seen in other parts of the world (e.g. in some European countries, where gp’s and pharmacists are considered co-responsible for pharmacotherapeutic treatment). This aspect warrants mentioning and some discussion.
  2. The central organization for guiding workplace learning appears to be the SLON (lines 191-195). A short description of the membership (numbers, coverage, distribution over the country) could help in supplying credibility to the present framework. Only 21 SLON-members commented during phase-2 of the study (line 271) and this may raise some questions about representativeness and potential bias in the results. Please comment shortly.
  3. Was saturation of opinions reached during phase-3, when apparently 16 selected individuals were interviewed (lines 333-355)?
  4. Reference to table-1 should be made in the text (around line 273).
  5. Figure 1 is a nice summary of the framework. Personally, I don’t think that it is necessary to attach the complete description as an Appendix, supposing that the framework will be available from the GPSA-website.

Author Response

This manuscript describes the development of the GPCLE-framework, based on an earlier version (the BPCLE-framework). The framework describes desirable elements and objectives for effective workplace learning environment. As such it can serve as a guideline for analysing and optimizing such learning environments.

Thank you for these positive comments.

The manuscript is well-written and describes the steps in developing the framework in a clear way. My main suggestion for improving the manuscript is that it should be made clear that the framework, and the way it was established, is relevant for the Australian context only. Aspects of the framework may be generic and/or applicable to other parts of the world, but in my opinion the framework as such cannot be applied elsewhere in unmodified form due to local characteristics of community practices and roles of health care professionals. I therefore suggest that the Title is completed by adding “… in Australia” (at the end of the Title).

Thank you. We added “:Australia” to the title. At p. 13 line 463, of the Discussion, we also suggest that this framework could be applied to other contexts, but we note it would require further validation, and we further note this in brief, in the limitations at the end of the Discussion. We have explained Australian general practice in the introduction such that readers can assess the degree of equivalence with their own context should they wish to validate this.

Apparently, a literature-based background paper was developed in phase 1 (lines 169-170), but the content of this background paper is not shared or published independently. It would have been interesting to summarize at least the main findings of this study in order to see in which elements the GPCLE-framework can be considered generic (wider application) and in which elements it is specific for the Australian situation. Would it be possible to include the main findings of the background paper in this manuscript, in summarized form or table-format? Otherwise, at least a short referral to non-Australian literature would be appreciated. Also, in the Discussion section a short comparison to literature from other contexts is welcomed (lines 428-438) to make the manuscript more interesting to a non-Australian audience.

Thank you for suggesting this. In response, we added to the Results, at p.6 line 289, a summary of the background paper including that we sourced international literature and at p.7 line 303, we included text about the principles that emerged.

In the Discussion, we also added references to other frameworks for global health systems development, and more clearly noted that there are no other existing frameworks in this field at p.13 line 463.

Other, minor remarks:

The range of learners in general practice settings (lines 60-62) seems not to include pharmacists (unless the fall in the category allied health). Are pharmacists excluded deliberately or as a consequence of local situations? This seems strange as a tendency to increasing inter-professional practice and education is seen in other parts of the world (e.g. in some European countries, where gp’s and pharmacists are considered co-responsible for pharmacotherapeutic treatment). This aspect warrants mentioning and some discussion.

Thank you, we have now addressed this. We note that inclusions in the term ‘allied health’ my vary by country so we judged it would be better to include ‘pharmacy, dentistry…’  to ensure interpretation consistent with our intent.

The central organization for guiding workplace learning appears to be the SLON (lines 191-195). A short description of the membership (numbers, coverage, distribution over the country) could help in supplying credibility to the present framework. Only 21 SLON-members commented during phase-2 of the study (line 271) and this may raise some questions about representativeness and potential bias in the results. Please comment shortly.

We added description of the SLON network including their number and distribution on p.5 line 214, to add to the material already outlined about this group at p.5 line 223.

Was saturation of opinions reached during phase-3, when apparently 16 selected individuals were interviewed (lines 333-355)?

Thank you for suggesting this. We added more information to indicate saturation and the consistency of responses and any divergent themes at p.10 line 393.

Reference to table-1 should be made in the text (around line 273).

Thank you for suggesting this.

On p.7 at line 322 and line 330 the mention of the table is now made where the material in this table is mentioned. This is called ‘Table 2’ as as part of the response to reviewers, a new Table 1 has been added to the paper (to summarise the four Phases of the project). All tables were renumbered as a result.

Figure 1 is a nice summary of the framework. Personally, I don’t think that it is necessary to attach the complete description as an Appendix, supposing that the framework will be available from the GPSA-website.

For ease of access, we decided that the whole framework could be supplied.